# A Determinantal Point Process Latent Variable Model for Inhibition in Neural Spiking Data

**Jasper Snoek***
Harvard University
jsnoek@seas.harvard.edu

**Ryan P. Adams**
Harvard University
rpa@seas.harvard.edu

**Richard S. Zemel**
University of Toronto
zemel@cs.toronto.edu

## Abstract

Point processes are popular models of neural spiking behavior as they provide a statistical distribution over temporal sequences of spikes and help to reveal the complexities underlying a series of recorded action potentials. However, the most common neural point process models, the Poisson process and the gamma renewal process, do not capture interactions and correlations that are critical to modeling populations of neurons. We develop a novel model based on a determinantal point process over latent embeddings of neurons that effectively captures and helps visualize complex inhibitory and competitive interaction. We show that this model is a natural extension of the popular generalized linear model to sets of interacting neurons. The model is extended to incorporate gain control or divisive normalization, and the modulation of neural spiking based on periodic phenomena. Applied to neural spike recordings from the rat hippocampus, we see that the model captures inhibitory relationships, a dichotomy of classes of neurons, and a periodic modulation by the theta rhythm known to be present in the data.

## 1 Introduction

Statistical models of neural spike recordings have greatly facilitated the study of both intra-neuron spiking behavior and the interaction between populations of neurons. Although these models are often not mechanistic by design, the analysis of their parameters fit to physiological data can help elucidate the underlying biological structure and causes behind neural activity. Point processes in particular are popular for modeling neural spiking behavior as they provide statistical distributions over temporal sequences of spikes and help to reveal the complexities underlying a series of noisy measured action potentials (see, e.g., Brown (2005)). Significant effort has been focused on addressing the inadequacies of the standard homogenous Poisson process to model the highly non-stationary stimulus-dependent spiking behavior of neurons. The generalized linear model (GLM) is a widely accepted extension for which the instantaneous spiking probability can be conditioned on spiking history or some external covariate. These models in general, however, do not incorporate the known complex instantaneous interactions between pairs or sets of neurons. Pillow et al. (2008) demonstrated how the incorporation of simple pairwise connections into the GLM can capture correlated spiking activity and result in a superior model of physiological data. Indeed, Schneidman et al. (2006) observe that even weak pairwise correlations are sufficient to explain much of the collective behavior of neural populations. In this paper, we develop a point process over spikes from collections of neurons that explicitly models anti-correlation to capture the inhibitive and competitive relationships known to exist between neurons throughout the brain.

Although the incorporation of pairwise inhibition in statistical models is challenging, we demonstrate how complex nonlinear pairwise inhibition between neurons can be modeled explicitly and tractably using a determinantal point process (DPP). As a starting point, we show how a collection of independent Poisson processes, which is easily extended to a collection of GLMs, can be jointly modeled in the context of a DPP. This is naturally extended to include dependencies between the individual processes and the resulting model is particularly well suited to capturing anti-correlation or inhibition. The Poisson spike rate of each neuron is used to model individual spiking behavior, while pairwise inhibition is introduced to model competition between neurons. The reader familiar with Markov random fields can consider the output of each generalized linear model in our approach to be analogous to a unary potential while the DPP captures pairwise interaction. Although inhibitory, negative pairwise potentials render the use of Markov random fields intractable in general; in contrast, the DPP provides a more tractable and elegant model of pairwise inhibition. Given neural spiking data from a collection of neurons and corresponding stimuli, we learn a latent embedding of neurons such that nearby neurons in the latent space inhibit one another as enforced by a DPP over the kernel between latent embeddings. Not only does this overcome a modeling shortcoming of standard point processes applied to spiking data but it provides an interpretable model for studying the inhibitive and competitive properties of sets of neurons. We demonstrate how divisive normalization is easily incorporated into our model and a learned periodic modulation of individual neuron spiking is added to model the influence on individual neurons of periodic phenomena such as theta or gamma rhythms.

The model is empirically validated in Section 4, first on three simulated examples to show the influence of its various components and then using spike recordings from a collection of neurons in the hippocampus of an awake behaving rat. We show that the model learns a latent embedding of neurons that is consistent with the previously observed inhibitory relationship between interneurons and pyramidal cells. The inferred periodic component of approximately 4 Hz is precisely the frequency of the theta rhythm observed in these data and its learned influence on individual neurons is again consistent with the dichotomy of neurons.

## 2 Background

### 2.1 Generalized Linear Models for Neuron Spiking

A standard starting point for modeling single neuron spiking data is the homogenous Poisson process, for which the instantaneous probability of spiking is determined by a scalar rate or intensity parameter. The generalized linear model (Brillinger, 1988; Chornoboy et al., 1988; Paninski, 2004; Truccolo et al., 2005) is a framework that extends this to allow inhomogeneity by conditioning the spike rate on a time varying external input or stimulus. Specifically, in the GLM the rate parameter results from applying a nonlinear warping (such as the exponential function) to a linear weighting of the inputs. Paninski (2004) showed that one can analyze recorded spike data by finding the maximum likelihood estimate of the parameters of the GLM, and thereby study the dependence of the spiking on external input. Truccolo et al. (2005) extended this to analyze the dependence of a neuron's spiking behavior on its past spiking history, ensemble activity and stimuli. Pillow et al. (2008) demonstrated that the model of individual neuron spiking activity was significantly improved by including coupling filters from other neurons with correlated spiking activity in the GLM. Although it is prevalent in the literature, there are fundamental limitations to the GLM's ability to model real neural spiking patterns. The GLM can not model the joint probability of multiple neurons spiking simultaneously and thus lacks a direct dependence between the spiking of multiple neurons. Instead, the coupled GLM relies on an assumption that pairs of neurons are conditionally independent given the previous time step. However, empirical evidence, from for example neural recordings from the rat hippocampus (Harris et al., 2003), suggests that one can better predict the spiking of an individual neuron by taking into account the simultaneous spiking of other neurons. In the following, we show how to express multiple GLMs as a determinantal point process, enabling complex inhibitory interactions between neurons. This new model enables a rich set of interactions between neurons and enables them to be embedded in an easily-visualized latent space.

### 2.2 Determinantal Point Processes

The determinantal point process is an elegant distribution over configurations of points in space that tractably models repulsive interactions. Many natural phenomena are DPP distributed including fermions in quantum mechanics and the eigenvalues of random matrices. For an in-depth survey,

see Hough et al. (2006); see Kulesza and Taskar (2012) for an overview of their development within machine learning. A point process provides a distribution over subsets of a space $\mathcal{S}$. A determinantal point process models the probability density (or mass function, as appropriate) for a subset of points, $S \subseteq \mathcal{S}$ as being proportional to the determinant of a corresponding positive semi-definite gram matrix $\mathbf{K}_S$, i.e., $p(S) \propto |\mathbf{K}_S|$. In the L-ensemble construction that we limit ourselves to here, this gram matrix arises from the application of a positive semi-definite kernel function to the set $S$. Kernel functions typically capture a notion of similarity and so the determinant is maximized when the similarity between points, represented as the entries in $\mathbf{K}_S$ is minimized. As the joint probability is higher when the points in $S$ are distant from one another, this encourages repulsion or *inhibition* between points. Intuitively, if one point $i$ is observed, then another point $j$ with high similarity, as captured by a large entry $[\mathbf{K}_S]_{ij}$ of $\mathbf{K}_S$, will become less likely to be observed under the model. It is important to clarify here that $\mathbf{K}_S$ can be any positive semi-definite matrix over some set of inputs corresponding to the points in the set, but it is not the empirical covariance between the points themselves. Conversely, $\mathbf{K}_S$ encodes a measure of anti-correlation between points in the process. Therefore, we refer hereafter to $\mathbf{K}_S$ as the kernel or gram matrix.

## 3 Methods

### 3.1 Modeling inter-Neuron Inhibition with Determinantal Point Processes

We are interested in modelling the spikes on $N$ neurons during an interval of time $\mathcal{T}$. We will assume that time has been discretized into $T$ bins of duration $\delta$. In our formulation here, we assume that all interaction across time occurs due to the GLM and that the determinantal point process only modulates the inter-neuron inhibition within a single time slice. This corresponds to a Poisson assumption for the marginal of each neuron taken by itself.

In our formulation, we associate each neuron, $n$, with a $D$-dimensional latent vector $\mathbf{y}_n \in \mathbb{R}^D$ and take our space to be the set of these vectors, i.e., $\mathcal{S} = \{\mathbf{y}_1, \mathbf{y}_2, \cdots, \mathbf{y}_N\}$. At a high level, we use an L-ensemble determinantal point process to model which neurons spike in time $t$ via a subset $S_t \subset \mathcal{S}$:

$$\Pr(S_t \mid \{\mathbf{y}_n\}_{n=1}^N) = \frac{|\mathbf{K}_{S_t}|}{|\mathbf{K}_{\mathcal{S}} + \boldsymbol{I}_N|}. \tag{1}$$

Here the entries of the matrix $\mathbf{K}_{\mathcal{S}}$ arise from a kernel function $k_\theta(\cdot, \cdot)$ applied to the values $\{\mathbf{y}_n\}_{n=1}^N$ so that $[\mathbf{K}_S]_{n,n'} = k_\theta(\mathbf{y}_n, \mathbf{y}_{n'})$. The kernel function, governed by hyperparameters $\theta$, measures the degree of dependence between two neurons as a function of their latent vectors. In our empirical analysis we choose a kernel function that measures this dependence based on the Euclidean distance between latent vectors such that neurons that are closer in the latent space will inhibit each other more. In the remainder of this section, we will expand this to add stimulus dependence.

As the determinant of a diagonal matrix is simply the product of the diagonal entries, when $\mathbf{K}_S$ is diagonal the DPP has the property that it is simply the joint probability of $N$ independent (discretized) Poisson processes. Thus in the case of independent neurons with Poisson spiking we can write $\mathbf{K}_S$ as a diagonal matrix where the diagonal entries are the individual Poisson intensity parameters, $\mathbf{K}_S = \mathrm{diag}(\lambda_1, \lambda_2, \cdots, \lambda_N)$. Through conditioning the diagonal elements on some external input, this elegant property allows us to express the joint probability of $N$ independent GLMs in the context of the DPP. This is the starting point of our model, which we will combine with a full covariance matrix over the latent variables to include interaction between neurons.

Following Zou and Adams (2012), we express the marginal preference for a neuron firing over others, thus including the neuron in the subset $S$, with a "prior kernel" that modulates the covariance. Assuming that $k_\theta(\mathbf{y}, \mathbf{y}) = 1$, this kernel has the form

$$[\mathbf{K}_S]_{n,n'} = k_\theta(\mathbf{y}_n, \mathbf{y}_{n'}) \delta \sqrt{\lambda_n} \sqrt{\lambda_{n'}}, \tag{2}$$

where $n, n' \in S$ and $\lambda_n$ is the intensity measure of the Poisson process for the individual spiking behavior of neuron $n$. We can use these intensities to modulate the DPP with a GLM by allowing the $\lambda_n$ to depend on a weighted time-varying stimulus. We denote the stimulus at time $t$ by a vector $\mathbf{x}_t \in \mathbb{R}^K$ and neuron-specific weights as $\mathbf{w}_n \in \mathbb{R}^K$, leading to instantaneous rates:

$$\lambda_n^{(t)} = \exp\{\mathbf{x}_t^\mathsf{T} \mathbf{w}_n\}. \tag{3}$$

This leads to a stimulus dependent kernel for the DPP L-ensemble:

$$[\mathbf{K}_S^{(t)}]_{n,n'} = k_\theta(\mathbf{y}_n, \mathbf{y}_{n'}) \, \delta \exp\left\{ \frac{1}{2} \mathbf{x}_t^\mathsf{T}(\mathbf{w}_n + \mathbf{w}_{n'}) \right\}. \tag{4}$$

It is convenient to denote the diagonal matrix $\mathbf{\Pi}^{(t)} = \mathrm{diag}(\sqrt{\lambda_1^{(t)}}, \sqrt{\lambda_2^{(t)}}, \cdots, \sqrt{\lambda_N^{(t)}})$, as well as the $S_t$-restricted submatrix $\mathbf{\Pi}_{S_t}^{(t)}$, where $S_t$ indexes the rows of $\mathbf{\Pi}$ corresponding to the subset of neurons that spiked at time $t$. We can now write the joint probability of the spike history as

$$\Pr(\{S_t\}_{t=1}^T \mid \{\mathbf{w}_n, \mathbf{y}_n\}_{n=1}^N, \{\mathbf{x}_t\}_{t=1}^T, \theta) = \prod_{t=1}^T \frac{|\delta \mathbf{\Pi}_{S_t}^{(t)} \mathbf{K}_{S_t} \mathbf{\Pi}_{S_t}^{(t)}|}{|\delta \mathbf{\Pi}_S^{(t)} \mathbf{K}_S \mathbf{\Pi}_S^{(t)} + \mathbf{I}_N|}. \tag{5}$$

The generalized linear model now modulates the marginal rates, while the determinantal point process induces inhibition. This is similar to unary versus pairwise potentials in a Markov random field. Note also that as the influence of the DPP goes to zero, $\mathbf{K}_S$ tends toward the identity matrix and the probability of neuron $n$ firing becomes (for $\delta \ll 1$) $\delta\lambda_n^{(t)}$, which recovers the basic GLM. The latent embeddings $\mathbf{y}_n$ and weights $\mathbf{w}_n$ can now be learned so that the appropriate balance is found between stimulus dependence and inhibition due to, e.g., overlapping receptive fields.

## 3.2  Learning

We learn the model parameters $\{\mathbf{w}_n, \mathbf{y}_n\}_{n=1}^N$ from data by maximizing the likelihood in Equation 5. This optimization is performed using stochastic gradient descent on mini-batches of time slices. The computational complexity of learning the model is asymptotically dominated by the cost of computing the determinants in the likelihood, which are $\mathcal{O}(N^3)$ in this model. This was not a limiting factor in this work, as we model a population of 31 neurons. Fitting this model for 31 neurons in Section 4.3 with approximately eighty thousand time bins requires approximately three hours using a single core of a typical desktop computer. The cubic scaling of determinants in this model will not be a realistic limiting factor until it is possible to simultaneously record from tens of thousands of neurons simultaneously. Nevertheless, at these extremes there are promising methods for scaling the DPP using low rank approximations of $\mathbf{K}_S$ (Affandi et al., 2013) or expressing them in the dual representation when using a linear covariance (Kulesza and Taskar, 2011).

## 3.3  Gain and Contrast Normalization

There is increasing evidence that neural responses are *normalized* or scaled by a common factor such as the summed activations across a pool of neurons (Carandini and Heeger, 2012). Many computational models of neural activity include divisive normalization as an important component (Wainwright et al., 2002). Such normalization can be captured in our model through scaling the individual neuron spiking rates by a stimulus-dependent multiplicative constant $\nu_t > 0$:

$$\Pr(S_t \mid \{\mathbf{w}_n, \mathbf{y}_n\}_{n=1}^N, \mathbf{x}_t, \theta, \nu_t) = \frac{|\nu_t \delta \mathbf{\Pi}_{S_t}^{(t)} \mathbf{K}_{S_t} \mathbf{\Pi}_{S_t}^{(t)}|}{|\nu_t \delta \mathbf{\Pi}_S^{(t)} \mathbf{K}_S \mathbf{\Pi}_S^{(t)} + \mathbf{I}_N|}, \tag{6}$$

where $\nu_t = \exp\{\mathbf{x}_t^\mathsf{T} \mathbf{w}_\nu\}$. We learn these parameters $\mathbf{w}_\nu$ jointly with the other model parameters.

## 3.4  Modeling the Influence of Periodic Phenomena

Neuronal spiking is known to be heavily influenced by periodic phenomena. For example, in our empirical analysis in Section 4.3 we apply the model to the spiking of neurons in the hippocampus of behaving rats. Csicsvari et al. (1999) observe that the theta rhythm plays a significant role in determining the spiking behavior of the neurons in these data, with neurons spiking in phase with the 4 Hz periodic signal. Thus, the firing patterns of neurons that fire in phase can be expected to be highly correlated while those which fire out of phase will be strongly anti-correlated. In order to incorporate the dependence on a periodic signal into our model, we add to $\lambda_n^{(t)}$ a periodic term that modulates the individual neuron spiking rates with a frequency $f$, a phase $\varphi$, and a neuron-specific amplitude or scaling factor $\rho_n$,

$$\lambda_n^{(t)} = \exp\left\{\mathbf{x}_t^\mathsf{T} \mathbf{w}_n + \rho_n \sin(f\,t + \varphi)\right\} \tag{7}$$

where $t$ is the time at which the spikes occurred. Note that if desired one can easily manipulate Equation 7 to have each of the neurons modulated by an individual frequency, $a_i$, and offset $b_i$. Alternatively, we can create a mixture of $J$ periodic components, modeling for example the influence of the theta and gamma rhythms, by adding a sum over components,

$$\lambda_n^{(t)} = \exp\left\{\mathbf{x}_t^\mathsf{T} \mathbf{w}_n + \sum_{j=1}^J \rho_{jn} \sin(f_j\,t + \varphi_j)\right\} \tag{8}$$

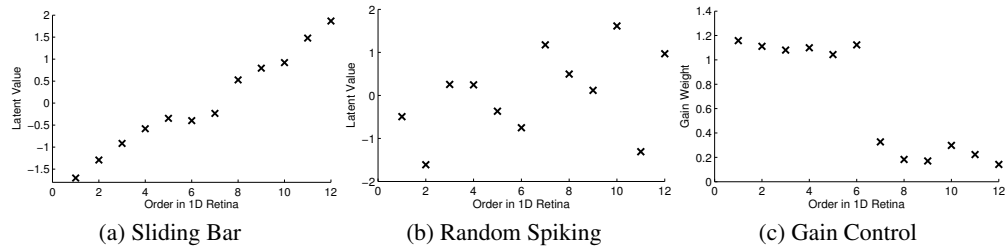

(a) Sliding Bar      (b) Random Spiking      (c) Gain Control

Figure 1: Results of the simulated moving bar experiment (1a) compared to independent spiking behavior (1b). Note that in 1a the model puts neighboring neurons within the unit length scale while it puts others at least one length scale apart. 1c demonstrates the *weights*, $\mathbf{w}_\nu$, of the gain component learned if up to 5x random gain is added to the stimulus at retina locations 6-12.

## 4 Experiments

In this section we present an empirical analysis of the model developed in this paper. We first evaluate the model on a set of simulated experiments to examine its ability to capture inhibition in the latent variables while learning the stimulus weights and gain normalization. We then train the model on recorded rat hippocampal data and evaluate its ability to capture the properties of groups of interacting neurons. In all experiments we compute $\mathbf{K}_S$ with the Matérn $5/2$ kernel (see Rasmussen and Williams (2006) for an overview) with a fixed unit length scale (which determines the overall scaling of the latent space).

### 4.1 Simulated Moving Bar

We first consider an example simulated problem where twelve neurons are configured in order along a one dimensional retinotopic map and evaluate the ability of the DPP to learn latent representations that reflect their inhibitive properties. Each neuron has a receptive field of a single pixel and the neurons are stimulated by a three pixel wide moving bar. The bar is slid one pixel at each time step from the first to last neuron, and this is repeated twenty times. Of the three neighboring neurons exposed to the bar, all receive high spike intensity but due to neural inhibition, only the middle one spikes. A small amount of random background stimulus is added as well, causing some neurons to spike without being stimulated by the moving bar. We train the DPP specified above on the resulting spike trains, using the stimulus of each neuron as the Poisson intensity measure and visualize the one-dimensional latent representation, $\mathbf{y}$, for each neuron. This is compared to the case where all neurons receive random stimulus and spike randomly and independently when the stimulus is above a threshold. The resulting learned latent values for the neurons are displayed in Figure 1. We see in Figure 1a that the DPP prefers neighboring neurons to be close in the latent space, because they compete when the moving bar stimulates them. To demonstrate the effect of the gain and contrast normalization we now add random gain of up to 5x to the stimulus only at retina locations 6-12 and retrain the model while learning the gain component. In Figure 1c we see that the model learns to use the gain component to normalize these inputs.

### 4.2 Digits Data

Now we use a second simulated experiment to examine the ability of the model to capture structure encoding inhibitory interactions in the latent representation while learning the stimulus dependent probability of spiking from data. This experiment includes thirty simulated neurons, each with a two dimensional latent representation, i.e., $N = 30$, $\mathbf{y}_n \in \mathbb{R}^2$. The stimuli are $16 \times 16$ images of handwritten digits from the MNIST data set, presented sequentially, one per "time slice". In the data, each of the thirty neurons is specialized to one digit class, with three neurons per digit. When a digit is presented, two neurons fire among the three: one that fires with probability one, and one of the remaining two fires with uniform probability. Thus, we expect three neurons to have strong probability of firing when the stimulus contains their preferred digit; however, one of the neurons does not spike due to competition with another neuron. We expect the model to learn this inhibition by moving the neurons close together in the latent space. Examining the learned stimulus weights and latent embeddings, shown in Figures 2a and 2b respectively, we see that this is indeed the case. This scenario highlights a major shortcoming of the coupled GLM. For each of the inhibitory

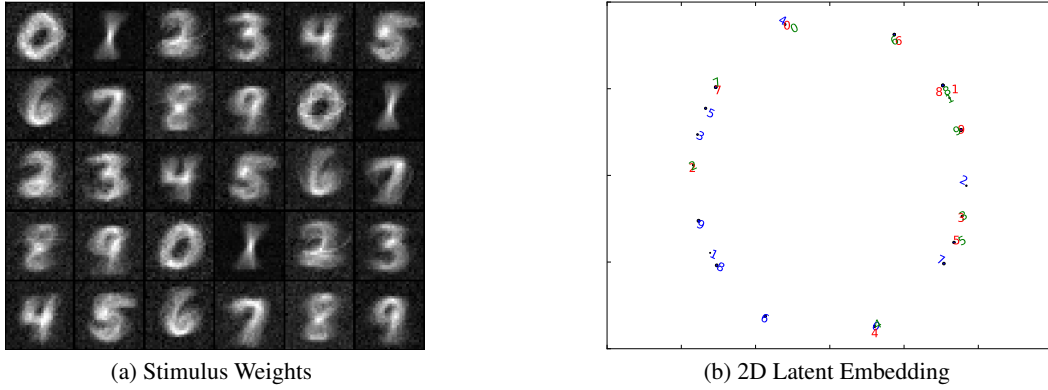

| (a) Stimulus Weights | (b) 2D Latent Embedding |

Figure 2: Results of the digits experiment. A visualization of the neuron specific weights $\mathbf{w}_n$ (2a) and latent embedding (2b) learned by the DPP. In (2b) each blue number indicates the position of the neuron that always fires for that specific digit, and the red and green numbers indicate the neurons that respond to that digit but inhibit each other. We observe in (2b) that inhibitory pairs of neurons, the red and green pairs, are placed extremely close to each other in the DPP's learned latent space while neurons that spike simultaneously (the blue and either red or green) are distant. This scenario emphasizes the benefit of having an inhibitory dependence between neurons. The coupled GLM can not model this scenario well because both neurons of the inhibitory pair receive strong stimulus but there is no indication from past spiking behavior which neuron will spike.

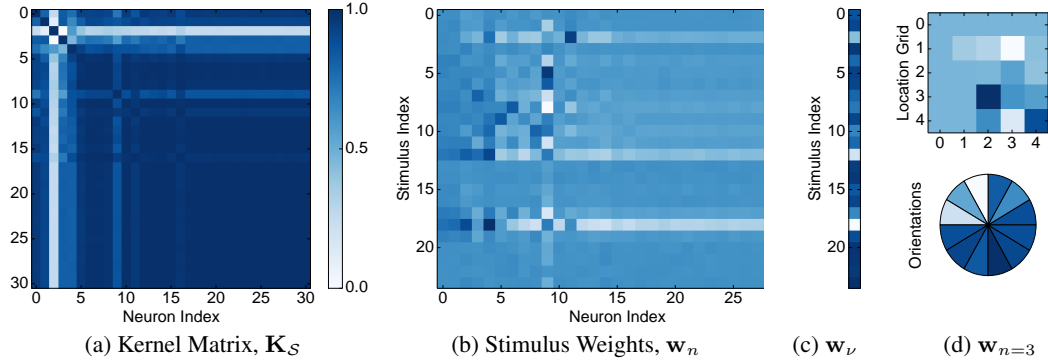

| (a) Kernel Matrix, $\mathbf{K}_S$ | (b) Stimulus Weights, $\mathbf{w}_n$ | (c) $\mathbf{w}_\nu$ | (d) $\mathbf{w}_{n=3}$ |

Figure 3: Visualizations of the parameters learned by the DPP on the Hippocampal data. Figure 3a shows a visualization of the kernel matrix $\mathbf{K}_S$. Dark colored entries of $\mathbf{K}_S$ indicate a strong pairwise inhibition while lighter ones indicate no inhibition. The low frequency neurons, pyramidal cells, are strongly anti-correlated which is consistent with the notion that they are inhibited by a common source such as an interneuron. Figure 3b shows the (normalized) weights, $\mathbf{w}_n$ learned from the stimulus feature vectors, which consist of concatenated location and orientation bins, to each neuron's Poisson spike rate $\lambda_n^{(t)}$. An interesting observation is that the two highest frequency neurons, interneurons, have little dependence on any particular stimulus and are strongly anti-correlated with a large group of low frequency pyramidal cells. 3c shows the weights, $\mathbf{w}_\nu$ to the gain control, $\nu$, and 3d shows a visualization of the stimulus weights for a single neuron $n = 3$ organized by location and orientation bins. In 3a and 3b the neurons are ordered by their firing rates. In 3d we see that the neuron is stimulated heavily by a specific location and orientation.

pairs of neurons, both will simultaneously receive strong stimulus but the conditional independence assumption will not hold; past spiking behavior can not indicate that only one can spike.

## 4.3 Hippocampus Data

As a final experiment, we empirically evaluate the proposed model on multichannel recordings from layer CA1 of the right dorsal hippocampus of awake behaving rats (Mizuseki et al., 2009; Csicsvari et al., 1999). The data consist of spikes recorded from 31 neurons across four shanks during open field tasks as well as the syncronized positions of two LEDs on the rat's head. The extracted positions and orientations of the rat's head are binned into twenty-five discrete location and twelve orientation bins which are input to the model as the stimuli. Approximately twenty seven minutes of spike recording data was divided into time slices of 20ms. The data are hypothesized to consist of spiking

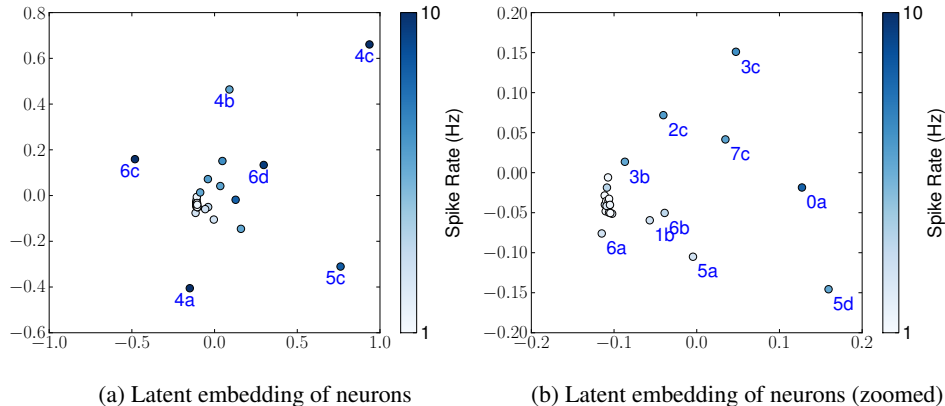

(a) Latent embedding of neurons      (b) Latent embedding of neurons (zoomed)

Figure 4: A visualization of the two dimensional latent embeddings, $\mathbf{y}_n$, learned for each neuron. Figure 4b shows 4a zoomed in on the middle of the figure. Each dot indicates the latent value of a neuron. The color of the dots represents the empirical spiking rate of the neuron, the number indicates the depth of the neuron according to its position along the shank - from 0 (shallow) to 7 (deep) - and the letter denotes which of four distinct shanks the neurons spiking was read from. We observe that the higher frequency interneurons are placed distant from each other but in a configuration such that they inhibit the low frequency pyramidal cells.

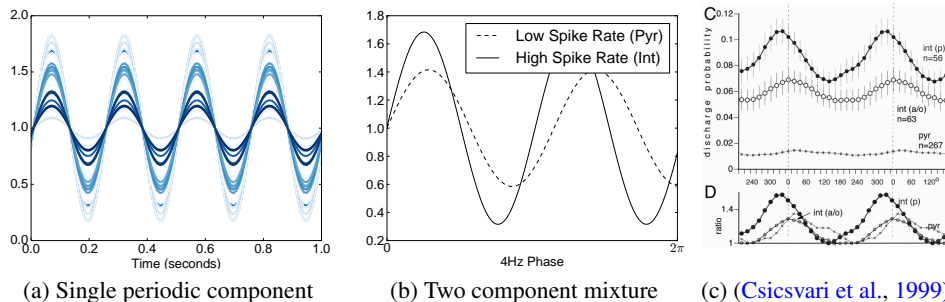

(a) Single periodic component      (b) Two component mixture      (c) (Csicsvari et al., 1999)

Figure 5: A visualization of the periodic component learned by our model. In 5a, the neurons share a single learned periodic frequency and offset but each learn an individual scaling factor $\rho_n$ and 5b shows the average influence of the two component mixture on the high and low spike rate neurons. In 5c we provide a reproduction from (Csicsvari et al., 1999) for comparison. In 5a the neurons are colored by firing rate from light (high) to dark (low). Note that the model learns a frequency that is consistent with the approximately 4 Hz theta rhythm and there is a dichotomy in the learned amplitudes, $\rho$, that is consistent with the influence of the theta rhythm on pyramidal cells and interneurons.

originating from two classes of neurons, pyramidal cells and interneurons (Csicsvari et al., 1999), which are largely separable by their firing rates. Csicsvari et al. (1999) found that interneurons fire at a rate of $14 \pm 1.43$ Hz and pyramidal cells at $1.4 \pm 0.01$ Hz. Interneurons are known to inhibit pyramidal cells, so we expect interesting inhibitory interactions and anti-correlated spiking between the pyramidal cells. In our qualitative analysis we visualize the the data by the firing rates of the neurons to see if the model learns this dichotomy.

Figures 3, 4 and 5a show visualizations of the parameters learned by the model with a single periodic component according to Equation 7. Figure 3 shows the kernel matrix $\mathbf{K}_{\mathcal{S}}$ corresponding to the latent embeddings in Figure 4 and the stimulus and gain control weights learned by the model. In Figure 4 we see the two dimensional embeddings, $\mathbf{y}_n$, learned for each neuron by the same model. In Figure 5 we see the periodic components learned for individual neurons on the hippocampal data according to Equation 7 when the frequency term $f$ and offset $\varphi$ are shared across neurons. However, the scaling terms $\rho_n$ are learned for each neuron, so the neurons can each determine the influence of the periodic component on their spiking behavior. Although the parameters are all randomly initialized at the start of learning, the single frequency signal learned is of approximately 4 Hz which is consistent with the theta rhyhtm that Mizuseki et al. (2009) empirically observed in these data. In Figures 5a and 5b we see that each neuron's amplitude component depends strongly

| Model | Valid Log Likelihood | Train Log Likelihood |
|---|---|---|
| Only Latent | $-3.79$ | $-3.68$ |
| Only Stimulus | $-3.17$ | $-3.29$ |
| Stimulus + Periodic + Latent | $-3.07$ | $-2.91$ |
| Stimulus + Gain + Periodic | $-3.04$ | $-2.92$ |
| Stimulus + Gain | $-2.95$ | $-2.84$ |
| Stimulus + Periodic + Gain + Latent | $-2.74$ | $-2.63$ |
| Stimulus + 2$\times$Periodic + Gain + Latent | $-2.07$ | $-1.96$ |

Table 1: Model log likelihood on the held out validation set and training set for various combinations of components. We found the algorithm to be extremely stable. Each model configuration was run 5 times with different random initializations and the variance of the results was within $10^{-8}$.

on the neuron's firing rate. This is also consistent with the observations of Csicsvari et al. (1999) that interneurons and pyramidal cells are modulated by the theta rhythm at different amplitudes. We find a strong similarity between the periodic influence learned by our two component model (5b) to that in the reproduced figure (5c) from Csicsvari et al. (1999).

In Table 1 we present the log likelihood of the training data and withheld validation data under variants of our model after learning the model parameters. The validation data consists of the last full minute of recording which is 3,000 consecutive 20ms time slices. We see that the likelihood of the validation data under our model increases as each additional component is added. Interestingly, adding a second component to the periodic mixture greatly increases the model log likelihood.

Finally, we conduct a leave-one-neuron out prediction experiment on the validation data to compare the proposed model to the coupled GLM. A spike is predicted if it increases the likelihood under the model and the accuracy is averaged over all neurons and time slices in the validation set. We compare GLMs with the periodic component, gain, stimulus and coupling filters to our DPP with the latent component. The models did not differ significantly in the correct prediction of when neurons would not spike - i.e. both were 99% correct. However, the DPP predicted 21% of spikes correctly while the GLM predicted only 5.5% correctly. This may be counterintuitive, as one may not expect a model for inhibitory interactions to improve prediction of when spikes do occur. However, the GLM predicts almost no spikes (483 spikes of a possible 92,969), possibly due to its inability to capture higher order inhibitory structure. As an example scenario, in a one-of-N neuron firing case the GLM may prefer to predict that nothing fires (rather than incorrectly predict multiple spikes) whereas the DPP can actually condition on the behavior of the other neurons to determine which neuron fired.

## 5   Conclusion

In this paper we presented a novel model for neural spiking data from populations of neurons that is designed to capture the inhibitory interactions between neurons. The model is empirically validated on simulated experiments and rat hippocampal neural spike recordings. In analysis of the model parameters fit to the hippocampus data, we see that it indeed learns known structure and interactions between neurons. The model is able to accurately capture the known interaction between a dichotomy of neurons and the learned frequency component reflects the true modulation of these neurons by the theta rhythm.

There are numerous possible extensions that would be interesting to explore. A defining feature of the DPP is an ability to model inhibitory relationships in a neural population; excitatory connections between neurons are modeled as through the lack of inhibition. Excitatory relationships could be modeled by incorporating an additional process, such as a Gaussian process, but integrating the two processes would require some care. Also, a limitation of the current approach is that time slices are modeled independently. Thus, neurons are not influenced by their own or others' spiking history. The DPP could be extended to include not only spikes from the current time slice but also neighboring time slices. This will present computational challenges, however, as the DPP scales with respect to the number of spikes. Finally, we see from Table 1 that the gain modulation and periodic component are essential to model the hippocampal data. An interesting alternative to the periodic modulation of individual neuron spiking probabilities would be to have the latent representation of neurons itself be modulated by a periodic component. This would thus change the inhibitory relationships to be a function of the theta rhythm, for example, rather than static in time.

## Footnotes

*Research was performed while at the University of Toronto.

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
