[Reviews · NeurIPS 2013]

Submitted by Assigned_Reviewer_1

This paper proposes determinantal point processes as a method to model inhibitory interactions in spike train data. The authors present a maximum likelihood approach based on stochastic gradient descent. This is an interesting idea that could potentially be a powerful non-factorial spike train model.

However, the presentation here seems a bit unfocused, and it’s not obvious to what extent DPPs will actually improve model accuracy. Since gain and periodic terms can easily be built into GLMs, it wasn’t obvious to me that framing the problem as a DPP was worth the trouble. Rather than looking at structure in the latent space, it seems more valuable to compare the utility of the DPP approach to other approaches.

GLMs with coupling between neurons, although not instantaneous, do capture correlations fairly well, and state-space models do even better with instantaneous correlations (cf Macke et al NIPS 2011). Several groups have also recently built models of instantaneous interactions using pseudo-likelihood approaches (Stevenson et al 2012 and Hanslinger et al Neural Comp 2013). How do these models, which might be considered more tractable, compare to the DPP approach?

Minor issues:
What kernel k_\theta was actually used? How important is it to use a kernel built on marginal preferences (as in Eq 2)?

It’s unclear to me from eq 6 how the gain control works here. In Carandini and Heeger’s framing normalization is a nonlinear operation on the rate of each neuron. Here \nu_t appears to be just a positive, stimulus-dependent scalar that affects the entire population. Is that right?

In section 2.1, the history isn’t quite accurate. GLMs were actually proposed to model spike trains much earlier (Brillinger 1988 and Chornoboy et al. 1988), and Harris et al. 2003 (rather than Truccolo et al.) was one of the first to really apply it to data.

In response to the rebuttal: the additional model comparison results are helpful. I'm a bit surprised that the DPP does so much better. It still may be useful to look into the GLM-like models that can be used to describe instantaneous coupling mentioned above.
Summary: This paper proposes determinantal point processes as a method to model inhibitory interactions in spike train data. This is an interesting idea that could potentially be a powerful non-factorial spike train model; however, the presentation here seems a bit unfocused.

Submitted by Assigned_Reviewer_6

This paper proposes the use of determinantal point processes (DPP) to model neural population activity. The authors show that the model can be fit to both simulated and real neural data.

A missing component of this paper is to show that it is an advance over existing methods, such as GLMs. In Section 4, the authors should compare the performance of the DPP and GLM on the same data, and show that DPP can outperform the GLM.

The authors state that one of the key benefits of DPP over GLM is that DPP can capture inhibitory interaction between neurons. Can't the GLM also capture inhibitory interactions via the coupling filters? If so, why is the way in which DPP captures inhibitory interactions superior? It would be helpful to provide 1-2 sentences of intuition about how equation (1) captures inhibitory interactions. Also, the authors emphasize the use of DPP for capturing inhibitory interactions - can't the DPP also capture excitatory interactions?

I had a hard time understanding Figures 2b, 3, and 4. It would be helpful if the authors would state what the reader is supposed to see. Is it possible to relate Figures 3 and 4 to the known identification of which neurons are excitatory vs. inhibitory?

On page 8, the authors state, "The model is able to accurately capture...dichotomy of neurons". I'm having trouble finding where in Section 4 this was shown.

In Section 3.2, it would be helpful to describe how to determine the dimensionality of y, and how to learn the kernel parameters (if present).
Summary: I think this work has good potential, but it seems underdeveloped at this point. The benefits of DPP over existing methods for modeling neural population activity needs to be clarified.

Submitted by Assigned_Reviewer_7

Summary:
--------

The authors apply Determinantal Point Processes (DPPs) to the spiking activity
of simultaneously recorded neurons. In addition to stimulus dependence, the
resulting model captures pairwise competitive interactions of neurons. The
authors apply the model to artificial data and hippocampal recordings.


Comments:
---------

1) The application of DPPs to neural recordings is novel to best of my
knowledge. The incorporation of instantaneous interactions into GLMs for
neurons (even coupled ones) such that the resulting model remains tractable is
an important open problem. Hence, I think the paper is timely and of interest
to the NIPS neuroscience audience.

2) The paper is clearly written and DPPs are introduced quite gently, resulting
in a very readable paper (exceptions below).

3) My main criticism is that the authors did not fully convince me that their
model is actually an appropriate one for multi-cell recordings. As far as I
know, most noise correlations that have been experimentally measured seem to be
positive (at least in cortex) and could therefore not be captured by this DPP
approach; please correct me if I'm wrong here. In any case, the authors
should have argued more thoroughly and given appropriate citations that the
scenario of exclusively competitive interactions is an important one in
multi-cell recordings.

4) Section 4.3, application to Hippocampus data: Unfortunately, this paragraph
does not fully convince me that the DPP based model is a good model for the
data. Are the noise correlations (computed from the data) between pairs of
excitatory / pairs of inhibitory neurons / pairs of exc-inhi really mostly
negative (emphasize here is on noise correlations as GLMs can capture stimulus
induced correlations)? The authors should describe the main result figures
FIG4 and FIG3(a) in greater detail: why is this latent embedding sensible / what
does this kernel matrix tell us about the data? (The additional space required
could be obtained by scaling back the experiments on artificial data.)
The fact that the method uncovers the theta oscillations is not very surprising
(as this is just the GLM part of the model) and could be described more
briefly. Table 1: The authors could have made a stronger point for the model if
the table included more pairs of models of the type
"GLM_component_1+...+GLM_component_n" and
"GLM_component_1+...+GLM_component_n+Latent", as this allows for a direct
comparison to figure out if adding the DPP part helps.

5) Section 3.3: In the model, the stimulus dependence of each neuron is already
captured by the weight vector $w_n$. Isn't the introduction of the of $w_\nu$
redundant?

6) The manuscript is missing a more in depth comparison between coupled GLMs
and the DPP approach, eg: coupled GLMs are lacking an instantaneous coupling
between the neurons, but this could be compensated for by binning the data on
finer time scales etc.


Minor Comments:
---------------

1) Either I don't fully understand the notation around eqn. 4 and 5 or it is is
rather sloppy: I guess $\Pi^{(t)}$ should be defined as
$diag(\sqrt{\lambda_1^{(t)}},\ldots)$, instead of the definition given in l157.
Furthermore in eqn 5, $K^{(t)}_{S_t}$ probably only contains the term $k_\theta$
and not the GLM part $\lambda_n^{(t)}$, in contrast to eqn 4. If this is
correct, then $K^{(t)}_{S_t}$ should be $K_{S_t}$ as the only time dependence
would be via ${S_t}$. Furthermore, shouldn't the normalizer in eqn 5 be
independent of $S_t$, i.e. contain eg $K_{\mathcal S}$ instead of $K_{S_t}$
(similar for $\Pi_{S_t}$)?

2) In the definition of $K_S$ in l131, it might be worthwhile to again emphasize
that $n,n' \in S$.
Summary: An interesting and timely paper that would benefit from extended
biological motivation for the proposed model and more detailed experiments.
Author Feedback

Author rebuttal: We thank the reviewers for their comments and valuable feedback on our paper. The reviewers raised some interesting questions, which we would like to address here.

Most of the feedback indicated that the reviewers desired more justification, both empirically and conceptually, of our model compared to the standard coupled GLM. We explain below the main theoretical difference and present an empirical comparison in a leave-one-neuron-out spike prediction experiment, which we believe significantly strengthens the paper.

We would like to emphasize that the DPP model developed in this paper is strictly a generalization of the GLM. In the absence of the latent component (when the kernel over latent values is the identity) it is equivalent to a collection of independent GLMs with Poisson outputs - to which one can add stimulus/coupling filters, etc. It has been an important open problem to model the dependence between neurons using the GLM and the coupled GLM was an attempt at doing this. There is, however, a major fundamental difference between our model and the coupled GLM. The DPP models the joint probability of a set of neurons firing in the same time window. The coupled GLM instead models them as conditionally independent. Under the DPP, one can easily evaluate the probability that a neuron fired in a time slice given that any other set of neurons fired. The coupled GLM models the probability of a neuron firing given which neurons fired in the previous time window. One resulting major difference in modeling ability is that the DPP can capture higher order interactions between sets of neurons, whereas the GLM can capture only pairwise causal interaction. For example, the DPP can model the case where only one neuron in a group fires despite them all receiving stimulus (we see this max-pooling like behavior e.g. in neuron populations with overlapping receptive fields in V1) - this can not be directly captured by the coupled GLM.

Prompted by the reviewers, we performed a leave-one-neuron-out spike prediction experiment on a withheld validation set from the hippocampal data. We compared directly to the coupled-GLM and in particular aimed to isolate the contribution of the DPP component. The results are interesting and more detail will be included in the paper (isolating each component) but for brevity we give a synopsis here. We would like to note that direct comparison to the standard coupled GLM was omitted in the submission due to its poor performance. The coupled GLM tends to behave poorly on prediction experiments with real data - an issue which we have corroborated in personal communication with redacted. We compared coupled GLMs with the periodic component, gain and stimulus to our DPP with the latent component. The models did not achieve significant differences in correctly predicting when neurons would not spike - i.e. both were ~99% correct. However, the DPP predicted 21% of actual spikes correctly while the GLM predicted only 5.1% correctly. This may be counterintuitive, as one may not expect a model for inhibitory interactions to be able to improve prediction of when spikes do occur. However, we believe that due to its inability to capture higher order inhibitory structure, the GLM simply learns to not predict any spikes. As an example scenario, in the one-of-N neuron firing case the GLM may prefer to predict that nothing fires (rather than incorrectly predict multiple spikes) whereas the DPP can actually condition on the behavior of the other neurons to determine which neuron fired. While we are not arguing that this is precisely the behavior of neurons in the hippocampus, it does appear from our results that there are complex inhibitory interactions in these data that the coupled GLM can not capture. This is precisely why we believe the model is appropriate for exploratory data analysis (and inference).

We would also like to emphasize (as we argued in the paper) that we are not suggesting that all interactions between neurons are negative and thus can be captured by the DPP. However, the DPP introduced is the only model that we are aware of that can model the joint distribution of a population of neurons while taking negative instantaneous interactions into account. Naturally, as a generative model, it can be combined with any of the models that can capture only positive interactions such as any GP-based model (e.g., GPFA) or MRFs, but we focus here on developing the inhibitory aspect. Inhibitory and competitive behavior is widely observed in neural data, e.g., interneurons acting on sets of pyramidal cells in the hippocampus (see citations below), so we believe this is a valuable contribution. Indeed the empirical analysis on real hippocampal data suggests that adding the inhibitory component to the model greatly improves prediction accuracy and model likelihood on withheld validation data, suggesting that competitive interactions may play a more important role in population responses than is currently believed.

In our empirical analysis we focused on the hippocampal data because of the known inhibitory behavior in the data. The dichotomy of neurons to which we refer is that of interneurons and pyramidal cells, the former known to inhibit the latter (see e.g. http://www.buzsakilab.com/content/PDFs/Royer2012.pdf, or Csicsvari et al. and Mizuseki et al. from the paper). In the latent space of the DPP (Figure 4), we observe this dichotomy as the high firing rate neurons, the interneurons, are placed far enough apart from each other to prevent inhibition but close enough to the low firing rate neurons, pyramidal cells, to cause inhibition. An interesting property is that the pyramidal cells are grouped together as they are probably inhibited collectively by the same interneurons. We will elucidate our observations more in the paper.

We agree that the descriptions of the figures in the paper were too brief. We will make these clearer in the paper.